# Self-Prepared Hyaluronic Acid/Alkaline Gelatin Composite with Nano-Hydroxyapatite and Bone Morphogenetic Protein for Cranial Bone Formation

**DOI:** 10.3390/ijms24021104

**Published:** 2023-01-06

**Authors:** Yuki Hachinohe, Masayuki Taira, Miki Hoshi, Daichi Yoshida, Wataru Hatakeyama, Tomofumi Sawada, Hisatomo Kondo

**Affiliations:** 1Department of Prosthodontics and Oral Implantology, School of Dentistry, Iwate Medical University, 19-1 Uchimaru, Morioka 020-8505, Iwate, Japan; 2Department of Biomedical Engineering, Iwate Medical University, 1-1-1 Idaidori, Yahaba-cho 028-3694, Iwate, Japan

**Keywords:** hyaluronic acid-based composite, alkaline-gelatin, chemical cross-link, ethylene glycol diglycidyl ether, nano-hydroxyapatite, bone morphogenic protein, bone formation, soft X-ray, histological observations

## Abstract

New bone-forming substitute materials are highly useful in dental implantology. The purpose of this study was to prepare cross-linked hyaluronic acid (cHLA)/cross-linked alkaline gelatin (cAG)/nano-hydroxyapatite (nHAp)/bone morphogenic protein (BMP) constructs; and evaluate their bone-forming capabilities in rat cranial bone defects. The cHLA and cAG liquids processed with an epoxy cross-linker were blended with a 3:1 volume ratio, followed by freeze-drying. The dry composites were further infiltrated with water containing nHAp only (BMP (−)) or with water containing nHAp and BMP (BMP (+)). Prepared wet constructs (BMP (−) and BMP (+)) were implanted in rat cranial bone defects, while defects only were also made, and animals were fed for 8 weeks, followed by subsequent soft X-ray measurements and histological observations. The X-ray results showed that BMP (+) constructs disappeared, though caused inward extension of peripherical bone from defect edges with an increase in length of approximately 24%, larger than those of BMP (−) constructs and defect only with approximately 17% and 8% increments, respectively (*p* < 0.05). Histological observations of BMP (+) construct samples clearly indicated active bone extension consisting of an array of island-like bones. It was concluded that cHLA/cAG/nHAp/BMP could be used as novel bone-substitute materials.

## 1. Introduction

In dental implantology and oral surgery, new bone-forming materials with increased bone-forming capabilities are expected to widen the shallow and narrow alveolar bones of patients, enabling bone drilling and subsequent implant insertion to be safely and effectively conducted [1,2].

Hyaluronic acid (HLA) is a natural polysaccharide constituting of D-glucuronic acid and N-acetylglucosamine [3] and is a component of the extracellular matrix of most connective tissues that display excellent biocompatibility within the human body [4]. Due to its chemical structure, HLA is a hydrophilic polymer that is soluble in water [3]. In vivo, HLA requires a chemical cross-link for a longer period, as hyaluronidase, abundantly distributed in the body, degrades uncross-linked HLA within 4 to 5 days [3]. For cross-linking, HLA has been chemically modified with hydrazide [5], amino or aldehyde functional groups [6], methacrylate groups [7], and thiol functional groups [8] to form stable cross-link networks [3,9]. HLA can also be directly cross-linked by an appropriate cross-linker, such as ethylene glycol diglycidyl ether (EGDE) [10,11].

Depending on the processing method, HLA materials can be prepared in the form of hydrogels, injectable gels, or sponges [3]. Often, HLA is employed as a gel material in the biomedical field [12,13,14,15], including an HLA gel with tri-calcium phosphate and bone morphogenic protein (BMP) later covered by collagen membrane in a tooth-extracted bone socket [16]. We have also previously reported that HLA sol–gel injectable materials with nano-hydroxyapatite (nHAp) and BMP cause ectopic and osseous bone formations [17,18]. Moreover, HLA can be used as a sponge or sheet for biomedical applications, such as artificial skin [10]. However, the use of HLA in sponge form for bone formation has been relatively limited. In a preliminary experiment, we succeeded in producing a cross-linked HLA (cHLA) sponge or sheet by chemical cross-linking of HLA with EDGE using the protocol developed by pioneers [10].

HLA itself is neither osteoinductive nor osteoconductive, while BMP is a strong osteoinductive growth factor [19]. Adding a growth factor and its carrier can render HLA-based materials osteoinductive and osteoconductive [19,20,21]. In this study, BMP-2 was employed as the United States Government Food and Drug Administration has only approved BMP-2 as a bone-inducing agent, even though BMP-4, BMP-7, and other BMP types have the potential to accelerate bone formation [22].

nHAp has been reported to be an excellent osteoconductive, bio-absorbable, and excellent carrier material, while larger hydroxy-apatite blocks and granules are more inert, less bio-absorbable, and less protein-adsorbed [23]. Further, Tabata et al. [24,25] reported that alkaline gelatin (AG) bound and slowly released BMP, and performed several studies for bone regeneration using BMP-impregnated AG [26,27,28]. As BMP carriers, nHAp and AG are useful. Moreover, their dual use in HLA-based composite seems meaningful; however, this approach has not been trialed in academic fields as of yet. We have also succeeded in the chemical cross-linking of AG with EDGE to form cross-linked AG (cAG).

Whilst two to three combinations selected from cHLA, cAG, nHAp, and BMP have been examined as bone-forming materials [29,30,31,32,33], there are few reports utilizing cHLA, cAG, nHAp, and BMP simultaneously as a quaternary composite [34] for bone formation studies.

Regarding in vitro culture tests of osteoblasts using nHAp-containing bio-materials, we already conducted culture tests of SaOS-2 osteoblastic cells on a nHAp-containing collagen sponge and confirmed that the composite was bio-compatible and accelerated osteogenic differentiation [35]. Moreover, there have been no reports of adverse biological effects with HLA and AG. As HLA and AG were main constituents of connective tissues such as collagen [4], cHLA/cAG/nHAp are also considered bio-compatible and osteoconductive. Holding materials for nHAp in bone substitute materials have wide varieties in material types including metals, polymers, ceramics, and composites, which can be either bioabsorbable or non-bioabsorbable [36]. Our cHLA/cAG was categorized as a naturally derived bioabsorbable polymer. Other holding materials include synthesized bioabsorbable polymer, namely, polylactic acid microfibers coupled with graphene/nHAp whiskers [37]. The prepared quaternary constructs also included BMP. It is widely known that BMP in the culture medium exerted a significant positive effect on osteogenic differentiation and secretion of bone matrix protein [38]. Although the in vitro culture studies were not carried out in this study, we assumed that the prepared quaternary construct was biocompatible and assisted in the growth and differentiation of cultured osteoblasts.

Therefore, the purposes of this investigation were as follows. Firstly, a HLA-based composite containing AG (cHLA/cAG) was prepared by chemical cross-linking of the HLA and AG using EDGE, separately and by mixing them together, followed by freeze-drying. The cHLA/cAG composites were characterized by scanning electron microscopy (SEM), Fourier-transform infrared spectroscopy (FTIR), and thermogravimetry (TG)/differential scanning calorimetry (DSC) thermal analyses. Secondly, the solution containing nHAp and protein (substitute for BMP) was infiltrated into cHLA/cAG composite to form cHLA/cAG/nHAp/protein constructs. Hydrolytic degradation/protein released tests were then conducted. The freeze-dried and wet cHLA/cAG/nHAp/protein (BMP substitute) samples were observed with SEM and confocal laser scanning microscopy, respectively. Thirdly, the bone-forming capabilities of rat critical-sized cranial bone defects of prepared cHLA/cAG/nHAp/BMP constructs were examined primarily with soft X-rays and histological observations, to determine whether the said constructs could be used as quaternary bone substitute materials in the future.

## 2. Results

### 2.1. Characterization of cHLA/cAG

#### 2.1.1. FTIR

Figure 1 shows FTIR charts of (a) cHLA/cAG, (b) cHLA, and (c) cAG, respectively. It became evident that three samples had similar functional groups of OH or NH at around 3300 cm^−1^, C-H at 2900 cm^−1^, and C=O at around 1080 cm^−1^. Amide peaks (-CO-NH-) were scales used to differentiate the materials examined. cAG had characteristic amide I and amide II peaks at 1650 cm^−1^ and 1560 cm^−1^, respectively, both of which cHLA/cAG possessed. cHLA lacked these amide peaks; however, cHLA had a strong C-O peak at 1080 cm^−1^ which cHLA/cAG also possessed, with cAG having a C-O peak at a minimum level. Overall, cHLA/cAG comprised all the major functional groups of cHLA and cAG.

Figure 2 indicates the FTIR charts of (a) cHLA/cAG, (b) HLA/AG, and (c) EDGE, respectively. It was judged that the peaks of cross-linked cHLA/cGA contained incremented characteristic peak elements of EDGE cross-linker which were super-imposed on un-crosslinked HLA/AG. EDGE had C-H peaks from 2790 to 3050 cm^−1^, which peaked at 2880 cm^−1^. cHLA/cAG had a slightly larger C-H peak (*1) than HLA/AG with a corresponding peak (*2). EDGE also possessed large C-O peak from 1050 to 1200 cm^−1^ which peaked at 1110 cm^−1^. cHLA/cAG had a narrowly larger C-O peak (*3) than HLA/AG with a corresponding peak (*4). It was considered that cHLA/cAG was produced by cross-linking of HLA/AG with EDGE.

#### 2.1.2. (TG)/DSC Thermal Analyses

Figure 3 (at the top and bottom) indicates TG and DSC thermal analysis results of (a) cHLA/cAG, (b) cHLA, and (c) cAG, respectively. From the TG curves (Figure 3 top side), the weight loss of cHLA became extensive when the temperature increased to more than 210 °C. The weight drop accompanied by subsequent temperature rise was initially relatively steep and continued up to 480 °C. The weight loss of cAG with incremental temperature increases started from a higher temperature of around 225 °C, and gradually and slowly continued up to 480 °C. The trend of weight decline of cHLA/cAG with increasing temperature was between those of cHLA and cAG, but more closely resembled that of cAG. From DSC curves (Figure 3 bottom), cHLA had a broad endothermic peak between 220 °C and 430 °C while both cAG and cHLA/cAG displayed similar small endothermic peaks between 210 °C and 300 °C. Overall, the thermal properties of cHLA/cAG were judged to be similar to those of cAG and vastly different from those of cHLA.

### 2.2. Characterization of cHLA/cAG/nHAp/Protein

#### 2.2.1. Hydrolytic Degradation and Protein Release Tests

Figure 4 shows the hydrolytic degradation/protein release test results of two constructs including (a) nHAp (+) constructs (i.e., cHLA/cAG/nHAp/bovine serum albumin (BSA)) and (b) nHAp (−) constructs (cHLA/cAG/BSA), expressed by the cumulative elution mass (%) graphs as a function of immersion period (week). It was observed that the protein was released with time from both constructs. The nHAp (−) construct specimens released more protein than the nHAp (+) construct counterparts. It also became apparent that the addition of nHAp suppressed protein release from the nHAp (+) constructs (cHLA/cAG/nHAp/BSA). There was a statistical difference in eluted protein amounts between nHAp (−) and nHAp (+) constructs at any immersion period (*p* < 0.05, calculated by the Mann–Whitney U test).

#### 2.2.2. Observation with SEM

Figure 5a–c show low-, middle-, and high-magnified SEM photomicrographs of cHLA/cAG composite, respectively. An interconnected porous structure with a pore size ranging from 10 to 70 μm was confirmed.

Figure 5d–f indicate low-, middle-, and high-magnified SEM photomicrographs of dried cHLA/cAG/nHAp/BSA constructs, respectively. On the high magnification image (Figure 5f), many aggregated nHAp particles with sizes ranging from 1 to 20 μm were seen on cHLA/cAG porous substrate. The nHAp particles appeared to be embedded in cHLA/cAG composite.

#### 2.2.3. Observation with Confocal Laser Microscopy

Figure 6a–c show the bright field, fluorescence, and overlay confocal laser microscopy images of nHAp mixed with type I collagen labeled with fluorescein isothiocyanate (FITC), respectively. It was confirmed that FITC-labeled type I collagen bound to many nHAp particles. Because the scan was performed at one height, nHAp particles without FITC fluorescence were also seen.

Figure 6d–f indicate the bright field, fluorescence, and overlay confocal laser microscopy images of cHLA/cAG/nHAp/FITC-labeled type I collagen, respectively. The sample swelled and was virtually floating on the glass slide at a certain thickness. It became evident that nHAp particles strongly attracted type I collagen with FITC within the construct. The surrounding cHLA/cAG area around nHAp particles tended to be covered by FITC-labeled type I collagen.

### 2.3. Animal Studies of cHLA/cAG/nHAp/BMP

#### 2.3.1. Soft X-ray Analyses

Figure 7 shows representative soft X-ray images of rat cranial bone defects 8 weeks after the operation with and without construct materials including (a) defects only, (b) defects with BMP (−) constructs (i.e., cHLA/cAG/nHAp) and (c) defects with BMP (+) constructs (i.e., cHLA/cAG/nHAp/BMP), respectively. Figure 7d–f indicate the same defects with the addition of dual circles to Figure 7a–c, respectively. The original defect corresponding zone was encircled by a solid line while the edge of the newly extended bone was depicted by a broken line. The difference in the area between zones defined by solid and broken lines was indicative of the area of the newly extended bone. In defect only, the defect deformed to the ellipse with the long diameter in the sagittal direction reaching 8 mm. This was stretched from the original defect length of 6 mm. Bone extension from the original bone defect was noticed on all three samples. However, the implantation of the BMP (+) construct accelerated bone extension the most, followed by the BMP (−) construct, while the defect only produced the least bone extension.

Figure 8 shows a tooled box graph of bone extension rates of (a) defects only, (b) BMP (−) constructs, and (c) BMP (+) constructs. Bone extension rates were determined by length-online measurements (n = 8 for each defect) on soft X-ray images of rat cranial bone defects 8 weeks after the operation, respectively (n = 8 for BMP (+) constructs, n = 6 each for BMP (−) constructs and defects only). Results clarified that BMP (+) constructs extended bone at the rat cranial bone defect the most, followed by BMP (−), while bone defects only extended bone the least. There was a statistical difference between any two combinations of defects only, BMP (−) constructs and BMP (+) constructs (*p* < 0.05, calculated with the Kruskal–Wallis test).

#### 2.3.2. Micro-Computed Tomography (Micro-CT) Observation

Figure 9 shows micro-CT image of the sagittal plane of four rat cranial defects filled with cHLA/cAG/nHAp/BMP constructs 8 weeks after the operation. The distance between the cranial defects was shortened to approximately 6 mm, compared with defects only (approximately 8 mm). The bone extension was active from the left side, and the elongated bone had a steeple configuration.

#### 2.3.3. Decalcified Tissue Histology

Figure 10 shows Hematoxylin and Eosin (HE)-stained histological images of rat cranial bone defects 8 weeks after the operation with and without construct materials: (a–c) defect only; (d–f) defect with BMP (−) construct (i.e., cHLA/cAG/nHAp); and (g–i) defect with BMP (+) construct (i.e., cHLA/cAG/nHAp/BMP). Figure 10b,c were enlarged images from X1 in Figure 10a and X2 in Figure 10b, respectively. Figure 10e,f were enlarged images from X3 in Figure 10d and from X4 in Figure 10e, respectively. Figure 10h,i were enlarged images from X5 in Figure 10g and X6 in Figure 10h, respectively. On all three samples, bone extension trends were obvious. The distances of rat cranial bone defects were largest for bone defects only, followed by BMP (−) constructs, and lowest for BMP (+) constructs. The remained materials were not found, and bone extension was a common characteristic of these histological observations.

#### 2.3.4. Non-Decalcified Tissue Histology

Figure 11a shows a Villanueva-stained image of a rat cranial bone defect 8 weeks after implantation of BMP (+) construct (i.e., cHLA/cAG/nHAp/BMP). It became evident that bone extension from the left side was extensive reaching approximately 2 mm in length. The extended bone consisted of an array of island-like bones. Figure 11b indicates a magnified calcein (CL)-fluorescent image of X1 in Figure 11a. New bone formation was very active in the last week of the 8-week period. Figure 11c–e are magnified tetracycline (TC)/CL-double fluorescent images of X2, X3, and X4 in Figure 11b, respectively. These enlarged images confirmed active bone formation in the last three weeks, with two distinct modes of bone growth from the ossification center (Figure 11c) and bone extension from pre-existing bone (Figure 11d,e).

## 3. Discussion

The cHLA/cAG composite sponge was produced by cross-linking HLA with EGDE, followed by cross-linking AG with EGDE, and blending these two cross-linked liquids, followed by freeze-drying. The obtained sols appeared to be well mixed at a molecular level. Two separate types of cross-linking with EGDE were required as HLA without -NH_2_ functional groups was cross-linked between -COOH functional groups at pH = 4 [10,11], and AG was primarily cross-linked between -NH_2_ functional groups at pH = 7 [39]. Both sols were dialyzed with distilled water to remove the residual cross-linker and maintain a neutral pH, subsequent mixing at a 3 to 1 ratio was then easily performed. This molecular blending method of once cross-linked polymers seemed to be relatively novel. EGDE was known as a safer cross-linker than glutaraldehyde [40], and, thus, used in this study. We avoided the carbodiimide/N-hydroxysuccinimide cross-linker [41] because it did not produce satisfactory cHLA/cAG composites in the preliminary experiment.

Material characterization studies by FTIR (Figure 1 and Figure 2) and TG/DSC thermal analyses (Figure 3) confirmed that cHLA/cAG composite was made up of cHLA and cAG compositions and possessed thermal properties, more resembling those of cAG, respectively. The large decline in weight and big DSC peak of cHLA might arise from physisorbed water liberation from cHLA which occurred when heated more than 200 °C [42]. Therefore, a genuine amount of cHLA-constituting substance in cHLA/cAG composite might be smaller even though it occupied three-quarters of the composite volume in the sol mixture stage. It was considered that cAG part of cHLA/cAG composite had a low volume with little water content but might have a larger actual weight proportion.

We measured the amount of protein eluted to PBS (−) solution from cAG part of cHLA/cAG/nHAp(±)/BSA construct samples using the protein assay kit (Figure 4). However, we did not monitor the degradation products of the cHLA part, mostly, polysaccharides [3,4]. The final protein release level from nHAp (−) constructs (cHLA/cAG/BMP) was about 42 wt% of the original dry weight, which was almost equal to the sum of cAG portion in cHLA/cAG composite and BSA applied. The total losses of the nHAp (−) constructs were visually confirmed 3 weeks after the immersion tests. Conversely, the protein release into the solution was retarded in the case of nHAp (+) constructs (cHLA/cAG/nHAp/BSA). This was due to nHAp retaining BSA protein for a certain period, absorbing the once-released surrounding protein, and suppressing hydrolytic degradation of the cHLA-based composite. Therefore, nHAp played dual roles as a protein carrier and as a slow releaser of retained protein. It is well known that nHAp absorbs both many positively and negatively charged proteins by counter-charged portions [43]. This protein could be a growth factor such as BMP. As for the relationship between BMP and BSA, the protein release test results obtained using BSA could apply to those of BMP. While the molecular weight of BMP was smaller, less than one-third of that of BSA [44], BMP had an isoelectric point (pI) of 6.5 to 8.5, neutral to slightly basic [45,46]. Protein bands of BSA separated by electrophoresis usually contain larger proteins with a pI of 4.7 to 5.1, slightly acidic [47]. The two pI points of BMP and BSA were not vastly different. It was speculated that BSA with a low pI, as a substitute for BMP, might effectively bind to cAG that possessed a base charge (pI = 8.2) by electrostatic attraction in this study [25]. Regarding the relationship between cAG and BMP, BMP was polarized at different locations with acidic (pI < 5) and base charge (pI > 8) [46]. It appeared that the portion with an acidic charge of BMP might be the binding site for AG.

From the SEM observations (Figure 5), it was confirmed that cHLA/cAG composite had inter-connected porous structures, which are advantageous for bone tissue engineering in size and morphology [48]. The addition of nHAp and BSA to cHLA/cAG composite resulted in homogeneous dispersion of aggregated nHAp particles on cHLA/cAG porous structure. Because a high vacuum was applied during sputtering, it was considered that nHAp aggregates were firmly embedded into the cHLA/cAG composite. BSA may contribute to binding between nHAp and cHLA/cAG composite. Moreover, the cHLA portion of cHLA/cAG composite might pull nHAp particles inside the substrate with water in the gel stage due to the strong hydrophilicity of HLA. 

It was also an important finding that nHAp strongly attracted protein (BSA) both alone and as an element of the polymeric composite as evidenced by confocal laser microscopic images (Figure 6). This result implies that nHAp can be a carrier of growth factors including BMP [18,19].

Soft X-ray images indicated that BMP (+) constructs (cHLA/cAG/nHAp/BMP) caused significant bone extension at rat cranial bone defects, larger than those of defects only and BMP (−) constructs (cHLA/cAG/nHAp) (Figure 7 and Figure 8). Sagittal micro-CT images of BMP (+) constructs (Figure 9) confirmed this bone extension. The extended bone did not have the full thickness of the pre-existing bone but it had a steeple configuration. This was regarded as an ongoing process (on the way to finally achieving full bone regeneration). It was also shown that the addition of nHAp to cHLA/cAG without BMP (BMP (−) constructs) also increased the bone formation rate, compared with that of defects only (*p* < 0.05). This was possibly due to the osteoconductive activity of nHAp [35]. The addition of BMP to BMP (−) constructs (i.e., BMP (+) constructs) further increased the bone extension rate, compared with that of BMP (−) constructs, due to the strong bone inductive activity of BMP (Figure 8).

HE images reconfirmed the findings obtained by soft X-ray analyses. Bone extension trends were found on all three samples including defects only, BMP (−) constructs, and BMP (+) constructs (Figure 10). The edges of newly formed bones (Figure 10c,f,i) displayed similar morphologies. The magnitude of the newly extended and formed bone was thought to be medium compared with other relevant studies [49,50,51].

Villanueva-stained and fluor-labeled images of a BMP (+) construct revealed more detailed morphological features of the newly extended bone (Figure 11). Newly regenerated bones consisted of an array of island-like small bones. These bones would be merged and united with the pre-existing bone, leading to full bone regeneration with time. These bones may stem from the intramembranous bone formation mechanism, at which mesenchymal stem cells are grown and differentiate into osteoblasts, creating bone and bone matrix [52,53].

In retrospect, the prepared BMP (+) constructs (i.e., cHLA/cAG/nHAp/BMP) was an early disappearance type growth factor impregnated scaffold [7]. This disappeared 8 weeks after implantation in cranial defects. Considering the protein-released tests (Figure 4), this may have only lasted for 4 to 5 weeks. In the body, enzymes, such as gelatinase, collagenase, and hyaluronidase accelerated the deterioration of cHLA/cAG composite [54,55]. From the early disappearance of the scaffold, it was considered that BMP was released relatively early as a fast burst type in this animal study, for a total duration of 8 weeks. The biodegradation process of the scaffold could be delayed by increasing the cross-linking levels of cHLA/cAG by several methods such as an increase in the cross-linking time, a selection of different cross-linkers, and the addition of a different cross-linking method [3,11]. Moreover, the degree of new bone formation could be increased by such an approach. Research in this field is highly valuable. It is possible to prepare a new late-disappearing type of scaffold with growth factors. 

Finally, it should be noted that the dual use of nHAp and AG as BMP carriers was useful for sustaining bone formation over a long time when cHLA was complexed with cAG. Furthermore, nHAp had a strong ability to absorb and slowly release proteins (Figure 4 and Figure 6), while cAG provided the frame of the scaffold and the role of a second BMP carrier. The volume matrix, cHLA provided a hydrophilic space-making role for bone tissue engineering. The combination of cHLA, cAG, nHAp, and BMP is a novel bone substitute in the future.

## 4. Materials and Methods

### 4.1. Materials

#### 4.1.1. cHLA/cAG Composite

Raw materials used for the production of cHLA/cAG composites were microorganism-derived HLA (HYALURONSAN HA-SHY, average molecular weight = 1,500,000–3,900,000, Kewpie Co., Tokyo, Japan), AG (G-2773P, Nitta Gelatin Co., Osaka, Japan), EGDE (Denacol EX-810; Nagase Chemtex, Osaka, Japan), sodium chloride (NaOH) (Tokyo Kasei Co., Tokyo, Japan), 1 N sodium hydroxide (NaOH) solution (Kanto Chemical Co., Tokyo, Japan), 1N hydrochloric acid (HCl) solution (Nacalai Tesque Co., Kyoto, Japan) and distilled water. 

#### 4.1.2. Addition of nHAp and BMP to cHLA/cAG Composite

The drug, material, and solution used for the addition to cHLA/cAG composites were BMP (BMP-2, R&D Systems, Catalog Number 355-BM, Minneapolis, MN, USA), nHAp with a mean diameter of 40 nm (nano-SHAp, SofSera, Tokyo, Japan), and a 4 mM HCl solution diluted from a 1N HCl solution with distilled water. The nHAp particles were autoclaved before mixing.

### 4.2. Methods

#### 4.2.1. Preparation of Biomaterials

##### Preparation of cHLA/cAG Composite

For the production of cHLA liquid, HLA powder (1.25 g) was dissolved into distilled water to form 2.5 wt% sol at 20 °C (Liquid A) (50 mL). The EGDE cross-linker (10 mL) was diluted by three with distilled water (20 mL), and the pH was adjusted to be four with NaOH and HCl solutions using a pH/ion meter (F-24; Horiba, Ltd., Kyoto, Japan) (Liquid B). Liquid A (30 mL) was mixed with the liquid B (10 mL) at a 3-to-1 ratio, followed by chemical cross-linking at 50 °C for 3 h. The liquid mixture was then cooled at 4 °C for 1-h. Figure 12 indicates two major cross-linking points of HLA, including -COOH and -OH functional groups. Figure 13a indicates the schematic of cross-linking reaction of HLA with EGDE using -COOH groups at pH four. The sol mixture was then purified using dialysis (Mw = 12,000–14,000, Code 3-25; Thermo Fisher Scientific, Waltham, MA, USA) against exchanged distilled water (1 L), three times for 3 days. The volume of the cHLA was increased by 3.5-fold relative to that of the original uncross-linked HLA.

For the preparation of cAG liquid, AG (5 g) was dissolved in distilled water (45 mL) at 37 °C at a concentration of 10 wt% (A liquid). The cross-linker (1 mL) was diluted by 20 with distilled water (19 mL). NaCl was then added to form a NaCl concentration of 20%; and the pH was adjusted to be seven using a pH meter with HCl and NaOH solutions (B liquid). The B liquid of 1 volume (2 mL) was mixed with the A liquid of 19 volume (38 mL), followed by chemical cross-linking at 4 °C for 72 h. Figure 13b indicates the schematic of the cross-linking reaction of AG with EGDE using NH_2_ groups at a pH of seven. The mixture was then purified using dialysis (Mw = 12,000–14,000, Code 3-25; Thermo Fisher Scientific, Waltham, MA, USA) against exchanged distilled water (1 L), three times for 3 days. The volume of the cAG was increased by 3.75-fold relative to that of the original uncross-linked AG.

The cHLA/cAG composite sponges were produced by mixing the cHLA solution with a cAG solution by a 3:1 volume ratio by pipetting and pouring in plastic plates (84 × 54 × 12 mm), frozen at −80 °C for 12 h, and freeze-dried (FD-5N; Eyela Co., Tokyo, Japan) for 24 h. The cut sponge was then filled into a stainless-steel metal die and compressed with a lead weight of 23.52 N (2.4 kgf) for 5 min to form a circular disk (25 mm in diameter and 1.5 mm thick). The disk was subsequently punched to produce specimens (6 mm in diameter and 1.5 mm thick). Small cHLA/cAG disks were sterilized using ethylene oxide gas and stored in a desiccator for animal experiments. Those without sterilization were used for other in vitro tests.

##### Addition of nHAp and BMP to cHLA/cAG Composite

BMP (50 μg) was reconstituted with a 4 mM HCl solution (0.25 mL in total) with 0.5 wt% bovine fetal albumin standard (fraction V) (Production no. DK59769, Thermo Scientific Pierce, Waltham, MA, USA) as adjuvant and distilled water (5 mL in total). The BMP-containing solution (1 mL) was mixed aseptically in the clean bench with nHAp (13.6 μg) with pipetting and kept at 4 °C for 3 h. The aliquot (50 μL) with nHAp (0.68 μg) and BMP (2.5 μg) was immersed in each cHLA/cAG disk for animal studies (BMP (+) constructs = cHLA/cAG/nHAp/BMP). Figure 14 indicates the photo of the cHLA/cAG disk on a 35 mm culture dish after pouring the aliquot with nHAp and BMP using a syringe. The construct without the addition of BMP was also prepared as a control (BMP (−) = cHLA/cAG/nHAp).

### 4.3. Characterization of Biomaterials

The cHLA and cAG solutions were separately prepared and freeze-dried after freezing for materialistic comparison studies with cHLA/cAG composite. The uncross-linked HLA/AG solution by a 3:1 volume ratio was also prepared.

#### 4.3.1. cHLA/cAG Composite

##### FTIR

FTIR equipped with an attenuated total reflectance attachment (Nicolet6700, Thermo Fisher Scientific, Waltham, MA, USA) (using a single reflection diamond) was used for solid samples to characterize organic functional groups in cHLA, cAG, and cHLA/cAG composites. A microscopic infrared spectrophotometer (Nicolet Continu μm, Thermo Fisher Scientific, Waltham, MA, USA) was employed in the transmission mode for liquid samples to determine the functional groups of HLA/AG and EDGE, allowing for a comprehensive assessment of the cross-linking reaction of HLA/AG (n = 1 each).

##### TG/DSC Thermal Analyses

TG/DSC was performed on each 1 mg sample (cHLA, cAG, and cHLA/cAG composite) (n = 1), using specialized equipment (STA409C, Netzsch, Selk, Germany) to ensure that the thermal stability of cHLA/cAG composite could be scaled with reference to those of cHLA and cAG. The experimental conditions for TG/DSC were as follows: atmospheric gas, nitrogen; gas flow rate (sample), 50 mL/min; gas flow rate (reference), 20 mL/min; temperature range, 20 °C to 480 °C; heating rate, 10 °C/min; sample holder, open aluminum crucible; reference, alumina (6.8 mg).

#### 4.3.2. cHLA/cAG/nHAp/Protein

Because of the cost and the problem of material availability, BMP was replaced with the two proteins mentioned below.

##### Hydrolytic Degradation and Protein Release Tests

Small amounts of bovine serum albumin standard (BSA) (2 mg/mL) (1 mL) (Thermo Scientific, Rockford, IL, USA) (50 μg) and nHAp (0.68 μg) were added to dry cHLA/cAG composite disk on a 35 mm culture dish and kept at 4 °C for 3 h (nHAp (+) = cHLA/cAG/nHAp/BSA). The disk without nHAp was also prepared as control (nHAp (−) = cHLA/cAG//BSA). The disks were then placed in 1.5 mL test tubes (n = 6 for both nHAp (+) and nHAp(−) constructs). Phosphate buffered saline solution (−) at a volume of 1 mL was made from PBS tablets (no. T900, Takara Bio, Kusatsu, Shiga, Japan) and added to the disk samples in tubes stored at 37 °C in a constant temperature bath for 3 weeks. The solution was collected 1, 2, and 3 weeks later and stored at −20 °C until measurements while new saline solutions (1 mL) were added to disks 1 and 2 weeks later. The quantities of BSA eluted in solution were measured using a Pierce BCA protein assay kit (Thermo Scientific, Rockford, IL, USA) with six samples containing two repetition measurements (n = 6 × 2) enabling the hydrolytic degradation/ protein release kinetics and the protein binding/releasing trend of cHLA/cAG/nHAp/BSA constructs to be visualized in a time-dependent fashion.

##### Observations with SEM and Confocal Laser Microscopy

A cHLA/cAG/nHAp/BSA (i.e., nHAp (+) construct) sponge was freeze-dried after freezing and coated with OsO_4_ using an OPC60A (Filgen, Nagoya, Japan). SEM (SU8010, Hitachi High-Tech Corp., Tokyo, Japan) was used at 15 kV to observe dried cHLA/cAG/nHAp/BSA and clarify the presence of nHAp particles on cHLA/cAG. The cHLA/cAG disk only sputtered with OsO_4_ and was also observed with SEM for morphological comparison.

Confocal laser microscopy (A1RHD25, Nikon Co., Tokyo, Japan) was used to observe wet nHAp/protein and cHLA/cAG/nHAp/protein. FITC-labeled bovine type I collagen (1 mg/mL) (no. 4001, Chondrex, Inc., Redmond, WA, USA) (0.5 mL) was mixed with nHAp (0.6 mg) containing PBS (−) ×2 buffered solution (0.5 mL) in a 1.5 mL microtube and held at 4 °C for 12 h. Half the solution with nHAp was centrifuged at 56× *g* (rotation radius = 50 mm, rotation speed = 1000 rpm) for 1 min. The supernatants were discarded, and the bottom pellets were resuspended in PBS (−) solution (300 μL). The solutions were then transferred to exclusive glass base dishes (27 mm glass in diameter) (code number 3970-035, IWAKI AGC Techno Glass Co., Tokyo, Japan), stood still for 1-h, and then the powders on the glass bottoms were observed. Another half solution was added to cHLA/cAG composite sponge (about 200 mg) without pressing on the glass dish and washed three times with phosphate-buffered saline (−) solution (1 mL). Subsequently, the final saline solution (500 μL) was added, enabling the identification of the wet cHLA/cAG/nHAp/protein. Measurements were taken at an excitation wavelength of 488 nm and an emission wavelength of 500–550 nm.

### 4.4. Animal Experiments

#### 4.4.1. Surgery

We used 20 male 10-week-old Wistar rats (CLEA Japan, Tokyo, Japan) weighing 340 ± 16 g. The rats were housed in separate cages (three rats per cage) and provided with a standard diet and water ad libitum. Anesthesia with a mixture of isoflurane (3% vol) and oxygen (0.5 L/min) gas generated by a carburetor (IV-ANE; Olympus, Tokyo, Japan) was used. The centers of the rat calvariae were shaved, sterilized with 10% povidone-iodine, and injected with a local anesthetic (0.2 mL, 2% lidocaine with 1:80,000 epinephrine). Then, full-thickness periosteum flaps were elevated, and bone defects were created using a trephine bur (6 mm diameter; Implant Re Drill System, GC, Tokyo, Japan). Specimens from each disk sample (BMP (−) constructs (n = 6) and BMP (+) constructs (n = 8)) were implanted in the calvarial bone defects (Figure 15), whereas six holes were left empty (defect only) (n = 6). The defects were covered with self-prepared collagen membranes [56]. The flaps were repositioned and sutured using soft nylon (Softretch 4-0, GC, Tokyo, Japan). At 8 weeks after surgery, the rats were sacrificed using CO_2_ inhalation. The animal experiments were performed in accordance with the Guidelines for the Care and Use of Laboratory Animals and approved by the Institutional Ethics Committee of Iwate Medical University (19 March 2021; approval no. 02-035).

#### 4.4.2. Soft X-ray Measurements and Micro-CT

We performed soft X-ray analyses (M60; Softex, Tokyo, Japan) to evaluate new bone formation and extension at the cranial critical defects after implantation of the two wet constructs (BMP (−) (n = 6) and BMP (+) (n = 8)) as well as those of untreated defects (n = 6). Figure 16 illustrates top and cross-sectional views of the bone formation process at the rat cranial bone defects filled by cHLA/cAG/nHAp/BMP constructs. The original bone defect is indicated by solid lines; newly formed bone is indicated by broken lines and the area is in orange color. Soft X-ray analyses were performed to identify new bone formation. For determination of the bone extension rate, the length of the original defect zone (X_0_) and that of extended bone (X_1_ + X_2_) were measured, using ImageJ software (1.53k; National Institutes of Health, Bethesda, MD, USA). Bone extension rate was calculated by (X_1_ + X_2_)/X_0_ × 100 (%). The bone extension rate of each defect with and without constructs was judged by measuring two kinds of lengths (X_0_ and X_1_ + X_2_) along eight evenly spaced radial lines in each defect, averaged for BMP (−) and BMP (+) constructs and defects only.

We also evaluated the bone extension trend at the rat cranial defect by BMP (+) constructs by sagittal section images (n = 4) using a three-dimensional micro-CT system (Scan Xmate-L090H, Comscan Techno Co., Yokohama, Kanagawa, Japan) at the voltage of 76 kV, current of 100 μA, and slice thickness of 53 μm.

#### 4.4.3. Histological Analyses

##### Decalcified Tissue Samples

After feeding for 8 weeks, cranial bones with and without wet constructs (defects only, BMP (−) and BMP (+)) (n = 6 for each) were removed using a diamond saw (MC-201 Microcutter; Maruto, Tokyo, Japan) or scissors and fixed in 10% neutral buffered formaldehyde equivalent (Mildform; Wako Chemicals, Osaka, Japan) for 4 weeks at 4 °C. Bones were then decalcified in 0.5 wt% ethylene diamine tetra-acetate solution (Decalcifying solution B; Wako Chemicals, Osaka, Japan) for 4 weeks at 4 °C. Next, the bones were treated with graded alcohol and xylene and embedded in wax. The specimens in the wax were then cut into 5 μm sections using a microtome (IVS-410; Sakura Finetek, Tokyo, Japan). The sections were stained with hematoxylin and eosin (HE) and observed using fluorescence microscopy (All-in-one BZ-9000; Keyence, Osaka, Japan).

##### Non-Decalcified Tissue Samples

Fluorescent double staining was performed on two cranial bones out of eight rats implanted with BMP (+) constructs. Sequential labeling was performed to evaluate postoperative bone formation and remodeling. Rats received an intraperitoneal injection of tetracycline (TC) (oxytetracycline hydrochloride, Nacalai Tesque Co., Kyoto, Japan) (3 mg/100 g body weight) dissolved in physiological saline solution (Otsuka Pharmaceutic Co., Tokyo, Japan); 0.4 mL) at 5 and 6 weeks after the surgery, followed by injection of calcein (CL) (Wako Chemicals, Osaka Japan; 1 mg/100 g body weight) in 0.4 mL of physiological saline solution at 7 weeks and 7 weeks + 5 days (2 days before sacrifice) after surgery. The rat calvariae embedded with BMP (+) constructs were processed to observe the non-decalcified histological appearance. After fixation in 70% ethanol (99.5% pure, Junsei Chemical Co., Tokyo, Japan) at 4 °C for 1 week, the samples were dehydrated in a series of graded ethanol (1 day at each concentration) and placed in pure acetone (Kanto Chemical Co., Tokyo, Japan) for 24 h. The samples were then stained with Villanueva solution (222-01445; Wako Chemicals, Osaka, Japan), embedded in methyl methacrylate for 4 days, and chemically polymerized for 10 days. The non-decalcified resin blocks (almost 15 mm × 15 mm × 20 mm) were cut in the sagittal plane using a circular diamond cutter (MC-201 Microcutter; Maruto, Tokyo, Japan). The sections were attached to plastic slides, ground to a thickness of 20 μm using a precision lapping machine (ML-110N; Maruto, Tokyo, Japan), and manually polished. Histological analysis of the sections was performed using fluorescence microscopy (All-in-one BZ-9000; Keyence, Osaka, Japan)) for Villanueva-stained images and single CL-fluor-labeling, with confocal laser scan microscopy (C1si; Nikon Co., Tokyo, Japan) used for dual TC–CL-fluor-labeling analysis.

### 4.5. Statistical Analyses

Free statistical software (EZR version 1.55, Saitama Medical Center, Jichi Medical University, Saitama, Japan) [57] was used for nonparametric tests, such as the Kruskal–Wallis test and Mann–Whitney U test. The null hypothesis was rejected at *p* < 0.05.

## 5. Conclusions

We prepared cHLA/cAG composite sponges via separate chemical cross-linking of HLA and AG with EGDE, mixing, freeze-drying, and subsequently infiltrating dried composites with nHAp and BMP-containing water. FTIR and TG/DSC analyses were carried out on cHLA/cAG composites. Hydrolytic degradation/protein release tests and SEM/confocal laser microscopy observation were performed on cHLA/cAG/nHAp/protein. Implantation of cHLA/cAG/nHAp/BMP constructs into rat cranial bone defects for 8 weeks was conducted to examine their bone-forming capabilities.

Despite its limitations, we can draw several important conclusions from our study of cHLA/cAG/nHAp/BMP constructs. Firstly, cHLA/cAG consisted of components of cHLA and cAG, and their thermal properties resembled those of cAG. Secondly, hydrolytic degradation/protein release tests showed that prepared cHLA/cAG/nHAp/protein constructs were early disappearance-type scaffolds. Thirdly, SEM/confocal laser microscopy observations confirmed that nHAp particles adhered well to the porous cHLA/cAG matrix and absorbed abundant protein. Fourthly, cHLA/cAG/nHAp/BMP constructs could effectively form new bones and moderately extend the bones at rat cranial bone defects. Finally, further materialistic studies are required to develop methods to improve bone formation.

## Figures and Tables

**Figure 1 ijms-24-01104-f001:**
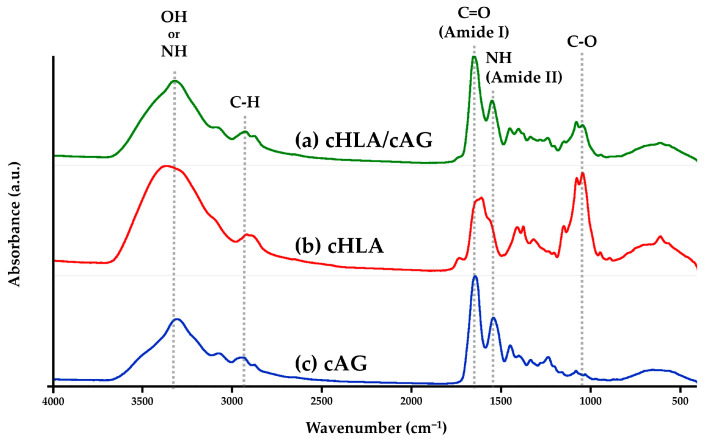
Fourier-transformed infrared spectroscopy (FTIR) charts of (**a**) cross-linked hyaluronic acid/cross-linked alkaline gelatin (cHLA/cAG), (**b**) cHLA, and (**c**) cAG. Note: Samples were cross-linked solids.

**Figure 2 ijms-24-01104-f002:**
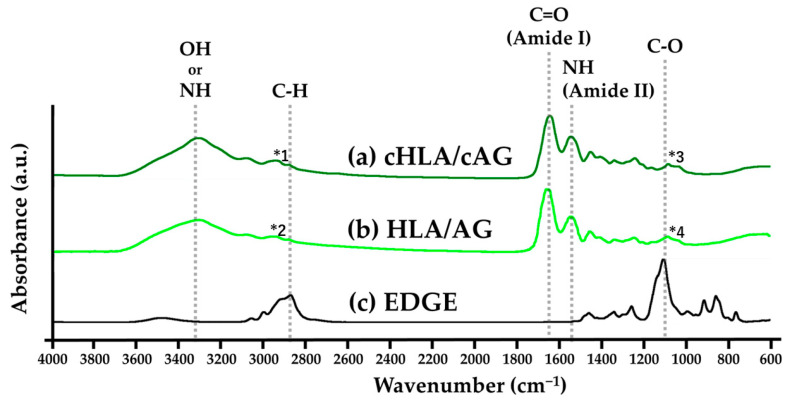
FTIR charts of (**a**) cHLA/cAG, (**b**) hyaluronic acid/alkaline gelatin (HLA/AG), and (**c**) ethylene glycol diglycidyl ether (EDGE). Note: cHLA/cAG was a cross-linked solid, HLA/AG was an uncross-linked liquid, and EDGE was a liquid cross-linker. The symbol * means the selected characteristic peak of FITR charts. The numbers are sequential order of appearance.

**Figure 3 ijms-24-01104-f003:**
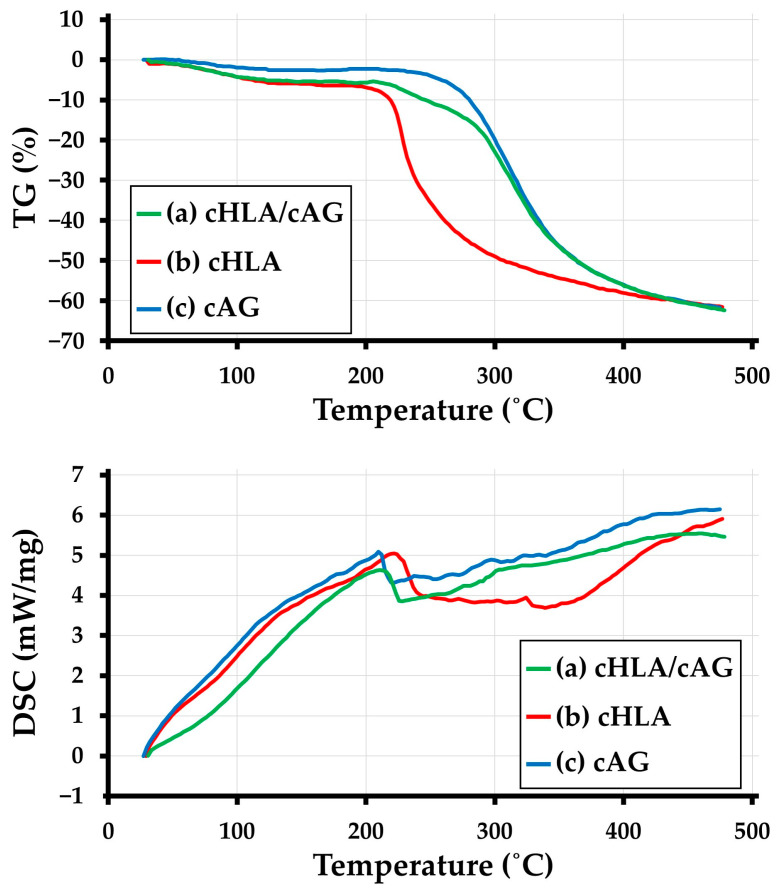
Thermogravimetry (TG) (Top) and differential scanning calorimetry (DSC) (bottom) thermal analyses results of (**a**) cHLA/cAG, (**b**) cHLA, and (**c**) cAG.

**Figure 4 ijms-24-01104-f004:**
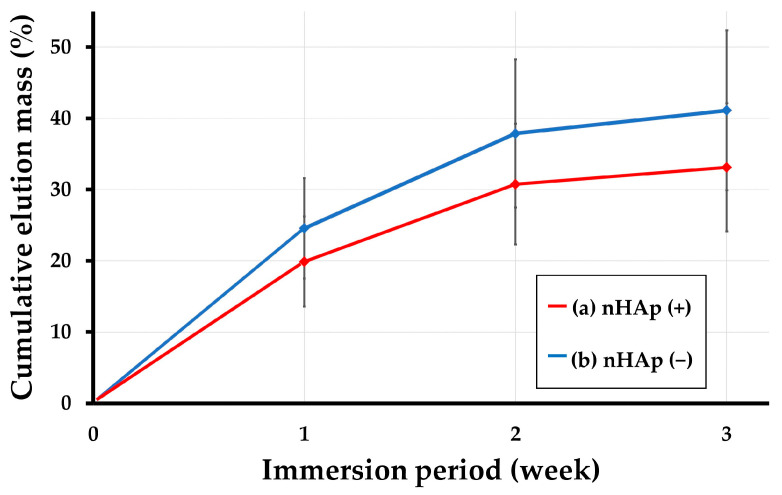
Hydrolytic degradation/protein release test results of two constructs including (**a**) nano-hydroxyapatite (nHAp) (+) constructs (i.e., cHLA/cAG/nHAp/bovine serum albumin (BSA)) and (**b**) nHAp (−) constructs (cHLA/cAG/BSA).

**Figure 5 ijms-24-01104-f005:**
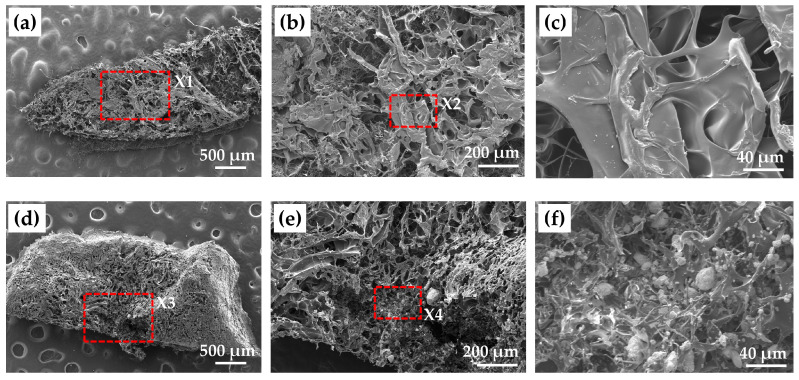
(**a**) Low-magnified scanning electron microscopy (SEM) image, (**b**) middle-magnified SEM image of the red square of X1 in (**a**), (**c**) high-magnified SEM image of the red square of X2 in (**b**) of dried cHLA/cAG composite; (**d**) low-magnified SEM image, (**e**) middle-magnified SEM image of the red square of X3 in (**d**), (**f**) high-magnified SEM image of the red square of X4 in (**e**) of dried cHLA/cAG/nHAp/BSA construct. Note: nHAp particles were seen on cHLA/cAG porous surface in (**d**–**f**).

**Figure 6 ijms-24-01104-f006:**
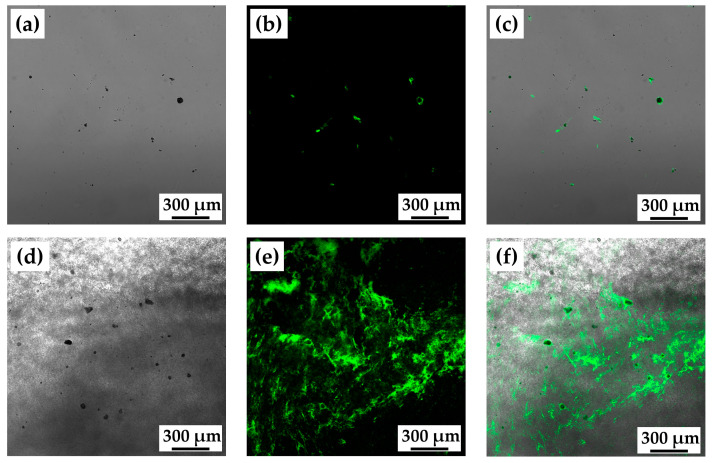
(**a**) The bright field, (**b**) fluorescence, and (**c**) overlay confocal laser microscopy images of nHAp mixed with fluorescein isothiocyanate (FITC)-labeled type I collagen; (**d**) the bright field, (**e**) fluorescence and (**f**) overlay confocal laser microscopy images of cHLA/cAG/nHAp/FITC-labeled type I collagen. Note: FITC-labeled type I collagen tended to highly bind to nHAp particles.

**Figure 7 ijms-24-01104-f007:**
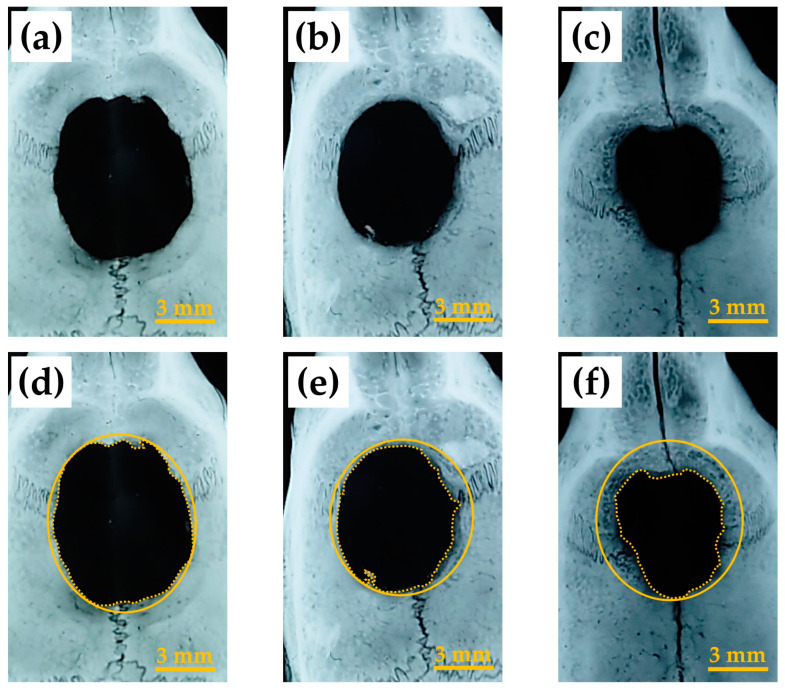
Representative soft X-ray image of rat cranial bone defects 8 weeks after the operation with and without construct materials including (**a**) defects only, (**b**) defects with BMP (−) construct (i.e., cHLA/cAG/nHAp) and (**c**) defects with BMP (+) construct (i.e., cHLA/cAG/nHAp/BMP); (**d**) same defect of (**a**) with two circles added, (**e**) that of (**b**), (**f**) that of (**c**). Note: In (**d**–**f**), solid line circles corresponded to the original defect while broken line circles indicated the edge of extending bone from defect peripheries.

**Figure 8 ijms-24-01104-f008:**
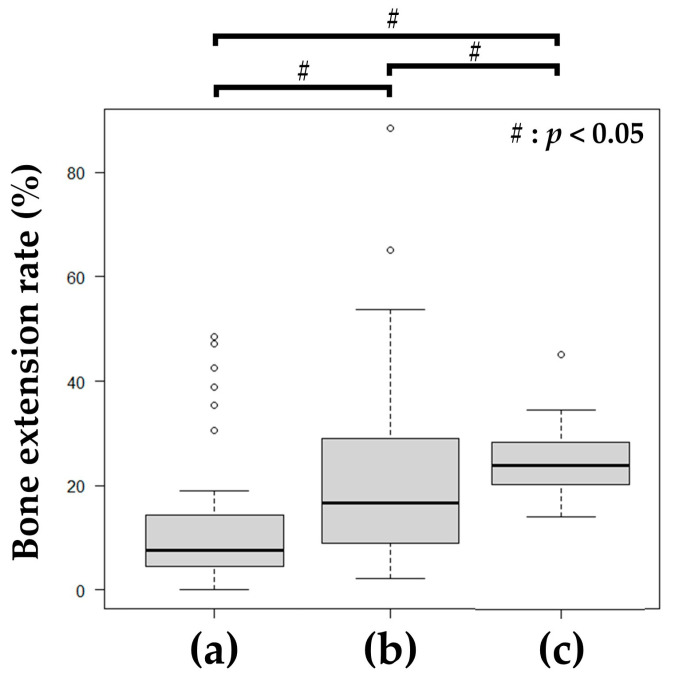
Tooled box graph of bone extension rates of (**a**) defects only, (**b**) BMP (−) constructs, and (**c**) BMP (+) constructs, determined by length-online measurements on soft X-ray images of rat cranial bone defects 8 weeks after the operation, respectively (n = 8 for BMP (+) constructs, n = 6 each for BMP (−) constructs and defects only). Note: Eight length-online measurements for each defect were performed. The statistics were carried out using the Kruskal–Wallis test.

**Figure 9 ijms-24-01104-f009:**
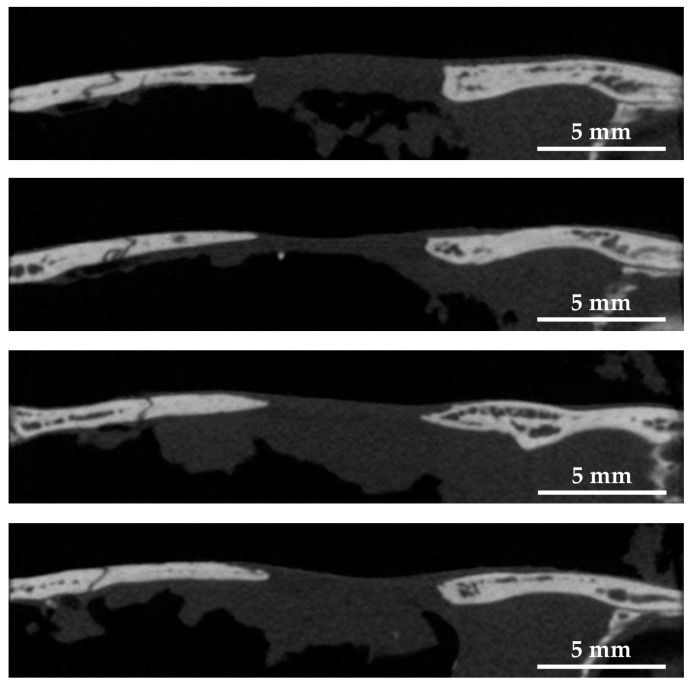
Micro-computed tomography (micro-CT) images of sagittal planes of four rat cranial bone defects filled with cHLA/cAG/nHAp/BMP constructs at 8 weeks after the operation. Note: The bone edge had a steeple configuration due to bone extension.

**Figure 10 ijms-24-01104-f010:**
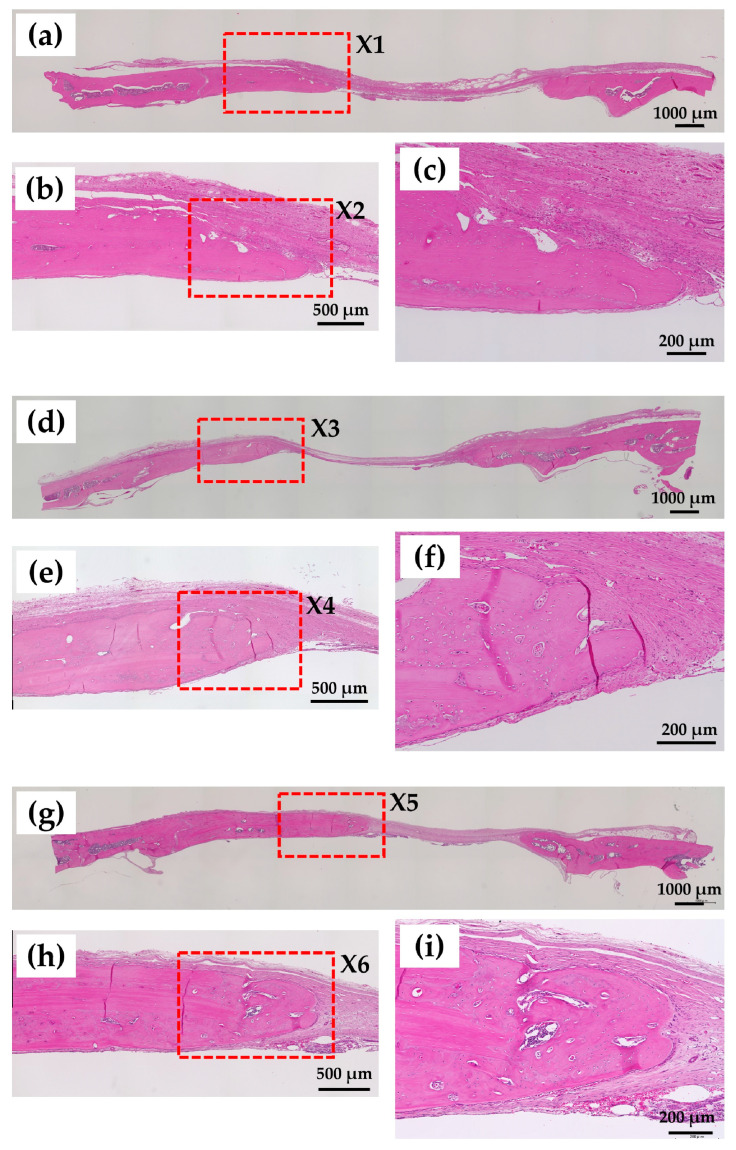
Hematoxylin and Eosin (HE)-stained histological images of rat cranial bone defects 8 weeks after the operation with and without construct materials including (**a**–**c**) defect only, (**d**–**f**) defect with BMP (−) construct (i.e., cHLA/cAG/nHAp) and (**g**–**i**) defect with BMP (+) construct (i.e., cHLA/cAG/nHAp/BMP). (**b**): enlarged from X1 in (**a**,). (**c**): enlarged from X2 in (**b**). (**e)**: enlarged from X3 in (**d**). (**f**): enlarged from X4 in (**e**). (**h**): enlarged from X5 in (**g**). (**i**): enlarged from X6 in (**h**).

**Figure 11 ijms-24-01104-f011:**
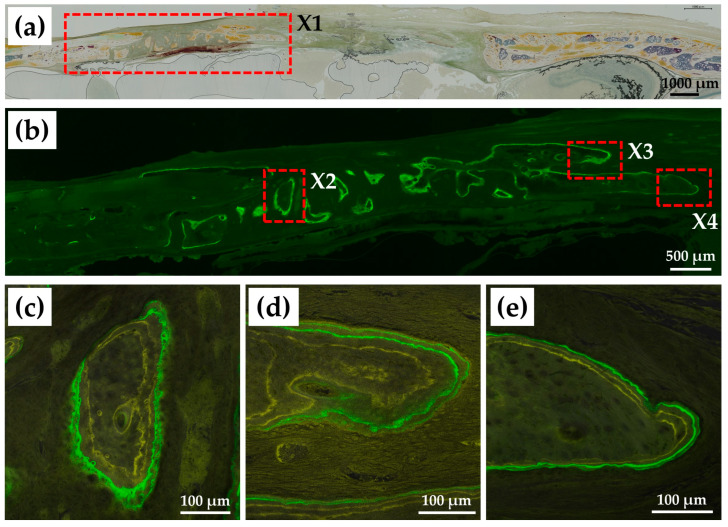
(**a**) Villanueva-stained image of a rat cranial bone defect 8 weeks after implantation of BMP (+) construct (i.e., cHLA/cAG/nHAp/BMP), (**b**) magnified calcein (CL)-fluorescent image of X1 in (**a**), (**c**) magnified tetracyclin (TC)–CL-dual fluorescent image of X2 in (**b**), (**d**) magnified TC–CL-dual fluorescent image of X3 in (**b**), (**e**) magnified TC–CL-dual fluorescent image of X4 in (**b**).

**Figure 12 ijms-24-01104-f012:**
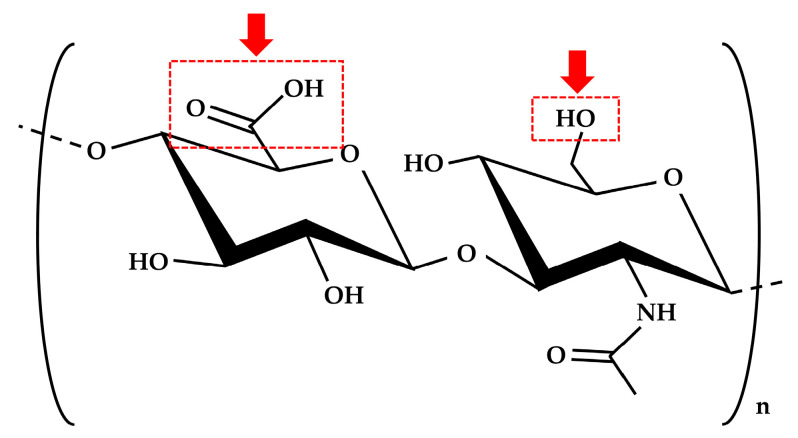
Two major cross-linking points of HLA, such as -COOH and -OH functional groups.

**Figure 13 ijms-24-01104-f013:**
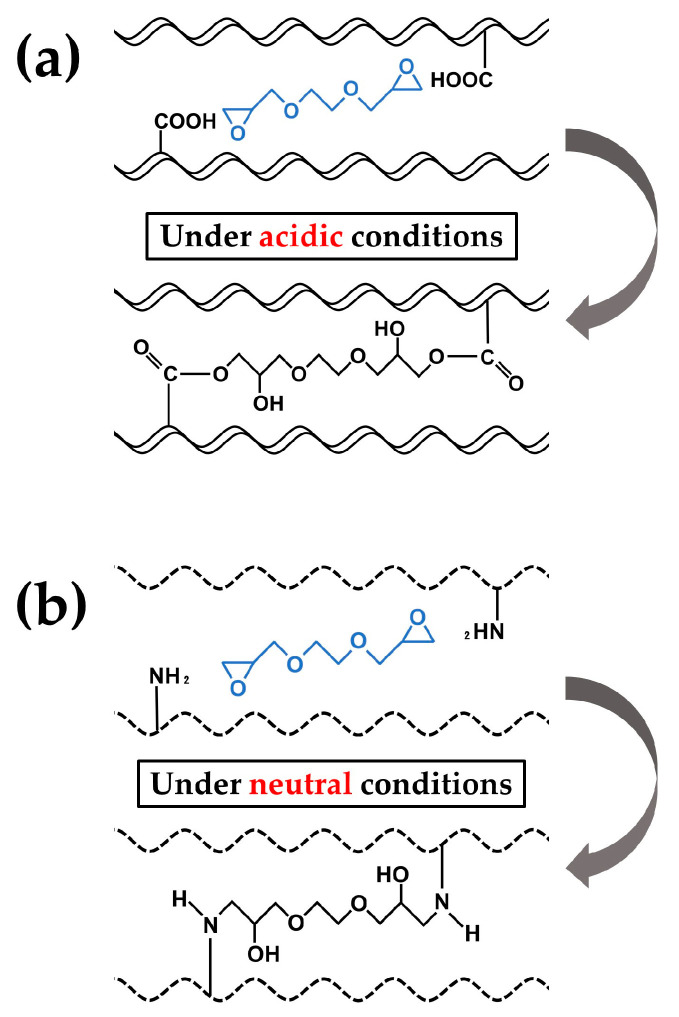
Schematic of cross-linking reactions of (**a**) HLA with EGDE using -COOH groups at pH = 4, and (**b**) AG with EGDE using NH_2_ groups at pH = 7.

**Figure 14 ijms-24-01104-f014:**
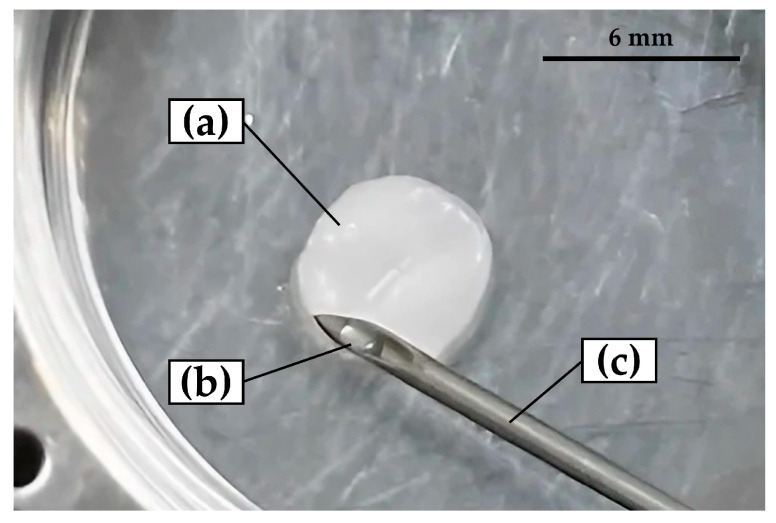
The photo of a cHLA/cAG disk on a 35 mm culture dish after pouring the aliquot with nHAp and BMP using a syringe. Note: (**a**) wet cHLA/cAG/nHAp/BMP construct, (**b**) residual aliquot, and (**c**) syringe used.

**Figure 15 ijms-24-01104-f015:**
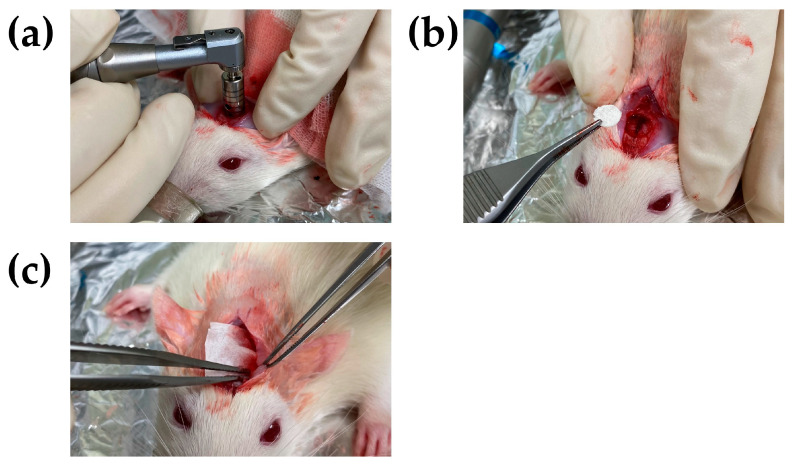
(**a**) Drilling a hole in rat cranial bone with trephine bur, (**b**) insertion of disk specimen in the cranial defect, (**c**) covering the defect with a self-prepared collagen membrane. Note: the membrane assisted in stopping bleeding, wound healing, preventing infection, fall prevention of the inserted construct, and bone formation, and disappeared within 8 weeks after insertion [56].

**Figure 16 ijms-24-01104-f016:**
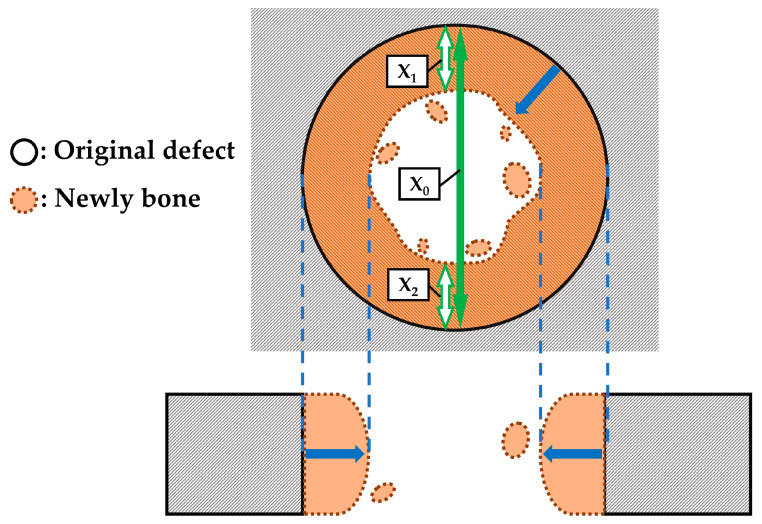
Illustration of top and cross-sectional views of the bone formation process at the rat cranial bone defects filled by cHLA/cAG/nHAp/BMP constructs. Note: Newly bone is depicted by zones in orange color. The edge of extending bone was shown by broken lines. The bone extension direction was indicated by the blue arrow. Bone extension rate was calculated by (X_1_ + X_2_)/X_0_ × 100 (%).

## Data Availability

All data are included in the manuscript.

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
