# Peer review of "Self-Prepared Hyaluronic Acid/Alkaline Gelatin Composite with Nano-Hydroxyapatite and Bone Morphogenetic Protein for Cranial Bone Formation"

_ijms, 2023, doi:10.3390/ijms24021104_

Round 1

Reviewer 1 Report

In this nicely conducted study, the authors fabricated cross-linked hyaluronic acid (cHLA)/cross-linked alkaline gelatin (cAG)/nano-hydroxyapatite (nHAp)/bone morphogenic protein (BMP) constructs and tested its osteogenesis effects with rat cranial bone defect model. The materials were degraded after 8 weeks and the bone growth extended from the borders of defects. The BMP+ group had more bone extension compared to the control group (BMP-), indicating the better osteogenesis effects of the BMP construct.

The authors should be commended for their open-minded to develop novel biomaterials for bone regeneration, the fundamental aspect of regeneration studies, that combined a couple of materials with their advantages to induce more bone formation.

There are just a few issues that may need to be considered:

1.     There is no in vitro experiment to test the biocompatibility and osteogenesis capacity of construct materials and it would be helpful if we can see the cell proliferation, differentiation and osteogenesis of the constructs.

2.     Authors may consider clarifying the reason for using BMP2 in this study rather than other types of BMP in the introduction part. 

Author Response

Thank you for your review, using your precious time.

I send you the replies to your comments by Word File.

With best regards

2022/12/30

Masayuki TAIRA

Reviewer 2 Report

The present work describes the synthesis of cross-linked hyaluronic acid/cross-linked alkaline gelatin/nano-hydroxyapatite/bone morphogenic protein constructs through a contrivable approach to effectively improve the thermal properties, hydrolytic degradation/protein release performance and new bones formation of the composite sponges. It is of interest to read this report, giving a great deal of characterization and performance evaluation. The paper are innovative, results are interesting and worth publishing, but the manuscript needs substantial improvements as follows:  

1. Some grammar errors are found in the manuscript, e.g., incorrect capital letters for the characterization methods: in line 90, 1080 cm-1 should be 1080 cm-1; in line 465, 500 μl should be 500 μL. Please check the full text.

2. The present title seems too long and has many key points, please check it clearly and concisely.

3. Figure 1, the FTIR curves only show major functional groups of cHLA and cAG. However, the groups’ changes of cross-linking cHLA/cAG should be also marked to support the schematic in Figure 12.

4. Figure 13, the resolution of the digital photo should be readjusted. Adding appropriate indication to point out different materials in the image.

5. The authors studied the nHAp composite scaffolds for bone regeneration in the manuscript, which might have wide applications in various fields. I would like to suggest the authors cite the following articles related to a similar composite scaffold with enhanced osteoblastic strength to enhance the literature: “Biomineralization of bone-like hydroxyapatite to upgrade the mechanical and osteoblastic performances of poly(lactic acid) scaffolds”, International Journal of Biological Macromolecules https://doi.org/10.1016/j.ijbiomac.2022.11.240.

Author Response

(The authors gave the same response as above.)
